# *Lacticaseibacillus casei* ATCC 393 Cannot Colonize the Gastrointestinal Tract of Crucian Carp

**DOI:** 10.3390/microorganisms9122547

**Published:** 2021-12-09

**Authors:** Hongyu Zhang, Xiyan Mu, Hongwei Wang, Haibo Wang, Hui Wang, Yingren Li, Yingchun Mu, Jinlong Song, Lei Xia

**Affiliations:** 1Fishery Resource and Environment Research Center, Chinese Academy of Fishery Sciences, Beijing 100141, China; zhanghy@cafs.ac.cn (H.Z.); muxiyan@cafs.ac.cn (X.M.); xuehu1110@aliyun.com (H.W.); wanghui@cafs.ac.cn (H.W.); liyr@cafs.ac.cn (Y.L.); 2Chinese Academy of Fishery Sciences, Beijing 100141, China; wanghongwei@cafs.ac.cn; 3Key Laboratory of Control of Quality and Safety for Aquatic Products (Ministry of Agriculture and Rural Affairs), Chinese Academy of Fishery Sciences, Beijing 100141, China

**Keywords:** *Lacticaseibacillus casei*, colonization, crucian carp, gastrointestinal tract, ^60^Co irradiation sterilization, transit marker, *Geobacillus stearothermophilus*, high-throughput sequencing

## Abstract

Lactic acid bacteria (LAB) are commonly applied to fish as a means of growth promotion and disease prevention. However, evidence regarding whether LAB colonize the gastrointestinal (GI) tract of fish remains sparse and controversial. Here, we investigated whether *Lacticaseibacillus casei* ATCC 393 (Lc) can colonize the GI tract of crucian carp. Sterile feed irradiated with ^60^Co was used to eliminate the influence of microbes, and 100% rearing water was renewed at 5-day intervals to reduce the fecal–oral circulation of microbes. The experiment lasted 47 days and was divided into three stages: the baseline period (21 days), the administration period (7 days: day −6 to 0) and the post-administration period (day 1 to 19). Control groups were fed a sterile basal diet during the whole experimental period, whereas treatment groups were fed with a mixed diet containing Lc (1 × 10^7^ cfu/g) and spore of *Geobacillus stearothermophilus* (Gs, 1 × 10^7^ cfu/g) during the administration period and a sterile basal diet during the baseline and post-administration periods. An improved and highly sensitive selective culture method (SCM) was employed in combination with a transit marker (a Gs spore) to monitor the elimination of Lc in the GI tract. The results showed that Lc (<2 cfu/gastrointestine) could not be detected in any of the fish sampled from the treatment group 7 days after the cessation of the mixed diet, whereas Gs could still be detected in seven out of nine fish at day 11 and could not be detected at all at day 15. Therefore, the elimination speed of Lc was faster than that of the transit marker. Furthermore, high-throughput sequencing analysis combined with SCM was used to reconfirm the elimination kinetics of Lc in the GI tract. The results show that the Lc in the crucian carp GI tract, despite being retained at low relative abundance from day 7 (0.11% ± 0.03%) to 21, was not viable. The experiments indicate that Lc ATCC 393 cannot colonize the GI tract of crucian carp, and the improved selective culture in combination with a transit marker represents a good method for studying LAB colonization of fish.

## 1. Introduction

Given the restrictions and prohibitions regarding the use of chemicals and antibiotics, there is an increasing demand for safe, cost-effective, and environmentally friendly feed supplements that possess exceptional benefits for farmed fish such as phytogenics, prebiotics and probiotics [1]. One of therapeutic benefits of probiotics are that they can colonize or temporally colonize gastrointestinal (GI) tract and thereby modulate the intestinal microbiota via competitive adherence and exclusion, resulting in the production of beneficial substances for the host [2,3]. Colonization is one of the most important characteristics when evaluating the application of probiotics in animal rearing. LAB are one of the most widely used and studied bacteria in aquaculture, but their colonization in the intestinal tract of fish remains highly debated. Tian et al. [4] stated that *Lacticaseibacillus casei* CC16 can colonize the intestines of common carp. Other papers have reported that *Pediococcus acidilactici* (Bactocell^®^, Lallemand Inc., Montreal, QC, Canada) [5], *Bacillus paralicheniformis* FA6 [6], *Lactiplantibacillus plantarum* G1 [7], *Lacticaseibacillus casei* ATCC 393 [8], *Latilactobacillus sakei* CLFP 202 [9], *Lactococcus lactis* CLFP 100 [9] and *Leuconostoc mesenteroides* CLFP 196 [9] can also colonize the GI tract of goldfish, grass carp, shabout fish and rainbow trout. However, some papers have shown that probiotic strains, including *Lactobacillus*, in the GI tract rapidly decreases following the withdrawal of supplementation [10,11,12,13,14,15,16], indicating their transient nature. Meanwhile, Ringø et al. [17] raised the following question: “Are probiotics permanently colonizing the GI tract?”.

Colonization was defined by Conway and Cohen as the indefinite persistence of a particular bacterial population without the reintroduction of that bacterium [18]. Most bacterial cells are transiently present in the GI tract of aquatic animals, with the continuous intrusion of microbes from water and food [19]. Commercial feed or homemade feed are usually unsterile except for specific pathogen free (SPF) or gnotobiotic animals [20]. Considering the widespread existence of lactic acid bacteria (LAB) and *Bacillus*, it is rational to speculate on their existence in aquafeed. The transient microbes in the GI tract enter water with feces and can then be reintroduced to that same GI tract. However, in probiotic colonization-related studies, little attention has been paid to the influence of microbes originating from feed and water, resulting in a conclusion that ignores the prerequisite for colonization, i.e., that it occurs “without the reintroduction of that bacterium”. In addition, the monitoring time for the persistence of probiotic microbes in the GI tract has often been insufficient, and there has been an absence of transit markers for evaluating the clearance time for transient microbes [21].

Colonization is a very important characteristic for screening additive strains and studying the mechanisms of probiotic action, but is associated with several significant challenges. First, the target bacteria being found in the water and diet can interfere with the colonization study. Second, lacking suitable methods for colonization study, some molecular methods such as 16S rRNA amplicon technology based on DNA samples cannot tell whether the bacteria are alive or dead. Third, once the probiotic supplementation has ceased, the proportion of the target strain may remain at a very low level [22], requiring a detection method with higher sensitivity for viable cells.

*L. casei* (Lc) is one of the species commonly used in aquaculture [4,17] and has shown some beneficial properties when applied to fish [23,24]. However, whether bacteria colonize the GI tract of fish has been unclear. To solve the issues above, the interfering microbes in feed and water were monitored and controlled, a transit marker was introduced, and an improved and highly sensitive selective culture method and high-throughput sequencing were both used to investigate whether *L. casei* can “truly” colonize the GI tract of crucian carp.

## 2. Materials and Methods

### 2.1. Bacteria Strains and Culture Condition

*Lacticaseibacillus casei* (Lc) ATCC 393 and *Geobacillus stearothermophilus* (Gs) ATCC 7953, were purchased from the China Center of Industrial Culture Collection and maintained with regular procedures.

Lc: The Lc strain was grown in MRS (De Man, Rogosa and Sharpe, Oxoid) broth at 37 °C overnight without agitation. The cells were harvested by centrifugation (5000× *g*, 5 min), resuspended in normal saline (0.85% (*w*/*v*) NaCl, pH 7.5) and adjusted to the necessary concentration.

Gs: The bacterial lawn grown on nutrient agar (NA, Aobox) supplemented with 18 μM/L MnSO_4_ at 57 °C for 4 days was harvested and washed twice with normal saline and then resuspended in normal saline. After inactivation vegetative cells incubated in a water bath at 90 °C for 30 min, the Gs spore suspension was centrifuged, washed twice with normal saline again, and then adjusted to the necessary concentration.

### 2.2. Experiment Diet

For the sterilized diet (basal diet), five commercial aquafeeds were sterilized by ^60^Co irradiation at 26.0 kGy, after which the efficacy of the sterilization was evaluated. The feed pellets with or without sterilization were homogenized and spread on nutrient agar and MRS agar with a pH of 5.4–5.5. The nutrient agar was incubated at 37 and 57 °C for 3 days to count the general heterotrophic bacteria and thermophiles, respectively. MRS agar was incubated at 37 °C for 3 days to count LAB. The colony number was counted to calculate the bacterial concentration in feed, and representative colonies with differing morphologies were selected for identification by 16S rRNA gene sequencing. Bacterial DNA was extracted using the TIANamp Bacteria DNA Kit (Tiangen) according to the manufacturer’s protocol. The DNA samples were submitted to the Rui Biotech, Inc. (Beijing, China) for PCR amplification and sequencing. The 16S ribosomal RNA gene from each sample were amplified and sequenced using the bacterial universal primer 27F (5′-AGAGTTTGATCCTGGCTCAG-3′) and 1492R (5′-TACGGCTACCTTGTTACGA CTT-3′). Then the 16S sequences alignments were performed using BLAST based on 16S ribosomal RNA sequences database of NCBI. Sterilized diet No. 2 (Beijing Fangteqi Feed Co., Ltd., Beijing, China, Table A1) was used in the experiment.

For the Mixed diet, Lc and Gs suspension were prepared and sprayed on the sterile basal feed to achieve a final concentration of 1 × 10^7^ cfu/g in Experiment 1. The final concentrations of Lc and Gs were 2 × 10^9^ cfu/g and 1 × 10^8^ cfu/g in Experiment 2, respectively. The experimental feed was air-dried in an oven for 10 min at 37 °C, and sealed and stored at 4 °C. The viable bacterial number in the feed was counted using the plate counting method at the beginning and end of the feeding experiments.

### 2.3. Experiment Design and Rearing Conditions

Two methods were used at two separate experimental phases. First, an improved and highly sensitive selective culture method (SCM) was established to compare the elimination kinetics between Lc and a transit marker (a Gs spore). Second, second-generation sequencing based on an 16S rRNA gene amplicon sequencing method (16S) was used to analyze the relative abundance of Lc and Gs. Meanwhile, the whole gastrointestines were sampled at the same time point, and their viable bacteria were monitored using the SCM. The flow chart of design of experiment see Figure A1.

Crucian carp (*Carassius auratus*) that weighed 20–40 g were obtained from the Beijing Longchi Aquaculture Farm. The fish were distributed into six separate glass aquariums (300 L) at a density of 24 fish per tank. Three glass aquariums were used for the treatment group (TG) and the others were used for the control group (CG). The study period was divided into three consecutive periods. First was the 21-day long, baseline period, during which the fish from both groups were fasted for 7 days and then acclimatized to the sterile pellet feed at 1.0–1.5% body weight once a day for 14 days. The last day of this period was defined as day −7. Next was the 7 day administration period (day −6 to day 0) and, finally, the post-administration period (19 days during Experiment 1 and 21 days during Experiment 2). During the administration period, the mixed diet was orally administered in both experiments for 7 days. The basal diet was used in all other periods, including the baseline period and the post-administration period. Meanwhile, the basal diet was used throughout the whole experiment in the control group. A total of nine fish with three in each tank were taken at days −7, 0, 7, 11, 15 and 19 during Experiment 1, whereas nine fish (six for the SCM and three for 16S) were collected at five time points during Experiment 2 (that is, days −7, 0, 7, 14 and 21).

During the baseline and post-administration periods, 100% of the water was renewed every 5 days in both experiments. Tap water was equilibrated to room temperature and aerated for 48 h before use. The physical parameters of the water were as follows: temperature 22–25 °C, pH 8.0–9.0, and dissolved oxygen > 6 mg/L.

### 2.4. Monitoring Lc and Thermophiles in Water

During the whole experimental period, 2 mL of water was sampled from the fish tanks every 3 days. A total of 1 mL water was spread on two MRS agar plates (pH 5.4–5.5, 500 μL on each plate), and the remaining 1 mL was spread on two nutrient agar plates (500 μL on each plate). MRS agar was incubated at 37 °C for 7 days. Nutrient agar was incubated at 57 °C for 2 days. The colonies were identified by 16S RNA gene sequencing.

### 2.5. Experiment 1: The Improved, Highly Sensitive Selective Culture Method

The pH of MRS medium was adjusted to 5.4–5.5 for the selective culture of Lc. The spore of Gs was used as the transit marker [21,25,26].

#### 2.5.1. Gastrointestine Homogenate Preparations

The fish were sacrificed at the sampling point, and close to the entire GI tract, from the esophagus to the anus, was aseptically removed. Then, an ice-cold normal saline solution was added to make a 10% (*w*/*w*) homogenate using a glass homogenizer. Meanwhile, the effect of the 10% GI tract homogenate on Lc and Gs and their respective media were evaluated as described below.

The Lc suspension was inoculated into the 10% GI tract homogenate of the crucian carp and normal saline at 1% (*v*/*v*) to a final concentration of 5 × 10^2^ cfu/mL. A 200 μL aliquot of homogenate containing Lc was spread on MRS agar with a pH of 5.4–5.5. A 200 μL aliquot of normal saline control containing Lc was spread on regular MRS agar. The plates were incubated at 37 °C for 7 days. Then, the colony number was counted to calculate the growth rate. The colony was identified at the species level by 16S RNA gene sequencing technology.

The Gs suspension was inoculated into the 10% GI tract homogenate of the crucian carp and the normal saline at 1% (*v*/*v*), achieving a final concentration of 1×10^3^ cfu/mL. Aliquots (100 μL) of homogenate and normal saline containing Gs were spread on the nutrient agar. The plates were incubated at 57 °C for 2 days, and the colony number was then counted to calculate the growth rate. The colony was identified at the species level by 16S rRNA gene sequencing technology.

The growth rate was assessed by Equation (1).
Growth rate = (the colony number of experiment group / the average colony number of control group) × 100%(1)

#### 2.5.2. Dynamics of Lc and the Transit Marker in the Gastrointestinal Tract

The GI tract was removed at the appropriate sample time point, and homogenate was prepared as described above (2.5.1); half was spread on nutrient agar (100 μL per plate), and the other half was spread on MRS agar with a pH of 5.4–5.5 (200 μL per plate). The detection limit for Lc and Gs was 2 cfu/gastrointestine. In cases where no LAB grew on the MRS agar, an additional six fish were sacrificed, and all GI tract homogenates were spread on MRS with a pH of 5.4–5.5 to reach a detection limit of 1 cfu/gastrointestine.

Generally, 20–30 plates are required for a 5 mL GI tract homogenate to reach a detection limit of 2 cfu/gastrointestine. The colonies were identified by microscopic examination and/or 16S rRNA gene sequencing. The sum of the Lc colony number for each plate was the total viable bacteria in the GI tract when all the GI tract homogenate was spread on the plate.

### 2.6. Experiment 2: 16S rRNA Gene Amplicon Sequencing Method (16S)

Sample Collection, DNA Extraction and Bioinformatic Analysis.

Of the nine samples (three fish per replicate) that were randomly selected from each group at each sample time point, six fish were monitored using the SCM as described above (2.5.2) and the other three fish were used for second-generation sequencing. The gastrointestinal contents were removed under sterile conditions. Bacterial DNA was extracted using the E.Z.N.A Mag-Bind Soil DNA Kit (Omega, Norcross, GA, USA). The DNA quality and concentrations were measured using a Qubit^®^3.0 spectrophotometer (Invitrogen, Waltham, MA, USA). The DNA samples were submitted to Sangon Biotech, Inc. (Shanghai, China) for PCR amplification and next-generation sequencing using an Illumina MiSeq platform. The primer sequences (341F (5′-CCTACACGACGCTCTTCCGATCTG(barcode) CCTACGGGNGGCWGCAG-3′) and 805R (5′-GACTGGAGTTCCTTGGCACCCGA GAATTCCAGACTACHVGGGTATCTAATCC-3′)), PCR cleanup, and sequencing were performed and a bioinformatic analysis was conducted as described in our previous study [27].

### 2.7. Data Analyses

Data were analyzed using *T*-test. A statistical analysis was performed using Microsoft Office Excel 2007(USA) with the level of significance set at *p* < 0.05.

## 3. Results

### 3.1. Effect of 100% Water Renewal on Interfering Bacteria

During the baseline period, no cultivable Lc or thermophiles were detected in the rearing water (<1 cfu/mL). During the administration period, 0–9 × 10^2^ cfu/mL of Lc and 0.1–8 × 10^3^ cfu/mL of Gs were detected in the rearing water. No Lc was detected following the cessation of bacterial supplementation and 100% water renewal up to the end of the experiments. Several Gs colonies were occasionally detected in the first week, whereas no Gs were detected after the second water renewal during the post-administration period.

### 3.2. Effect of Sterilizing the Feed with ^60^Co Irradiation

The bacterial content of the commercial aquafeed is shown in Table A2. There were general heterotrophic bacteria at 10^4^–10^6^ cfu/g of the commercial diet, LAB at 10^2^–10^4^ cfu/g and thermophiles at 10^2^–10^4^ cfu/g. Using 16S rRNA gene sequencing identification, it was found that the general heterotrophic bacteria were mainly species of the genera *Bacillus* (including *Bacillus licheniformis* and *Bacillus subtilis*), and others include *Enterobacter*, *Parabacillus*, *Pantoea**,* etc. The LAB were *Pediococcus*, *Enterococcus* and *Bacillus coagulans*. The thermophiles included mainly *Geo**bacillus*, *Parageo**bacillus*, and *Bacillus*. None of these bacteria were detected after ^60^Co irradiation sterilization.

Meanwhile, the concentration of Gs and Lc in the mixed diet did not attenuate at the end of either experiments (Table A3).

### 3.3. Selective Culture for LAB and Gs

The pH of MRS medium was adjusted to 5.4–5.5 for the selective culture of Lc. The MRS agar with a pH of 5.4–5.5 had high specificity for Lc growth, except for the occasional presence of some fungi and motile bacteria that failed to subculture in the rearing water and the gut at very low doses. There was no significant difference between the regular MRS and the 10% GI tract homogenate MRS (pH 5.4–5.5) (Figure 1). In other words, the improved MRS agar had a high specificity and sensitivity and was, thus, able to detect the LAB strains used in our study of the GI tract homogenate.

The growth rate of Gs at 57 °C was 83.78% ± 26.80% (Figure 2) when suspended in the 10% GI tract homogenate, which was slightly lower than that of the normal saline control. However, there were no significant differences between the two groups (*p* > 0.05).

### 3.4. The Concentration of Lc Changes in the GI Tract of Crucian Carp

The concentration of Lc and Gs in the GI tract decreased dramatically after the cessation of both bacteria supplements (Figure 3). In the first 3 days, the Lc concentration decreased from 2.6 × 10^5^ (5.43log) to 20.67 (1.32log) cfu/gastrointestine, and Lc could not be detected in the GI tracts of two out of nine fish. Seven days after the cessation of the mixed diet, Lc could not be detected in any of the sampled fish (< 2 cfu/gastrointestine), although Gs was remained detectable up to day 11 (7/9). As can be seen from Figure 3, Lc was eliminated from crucian carp gastrointestine faster than Gs.

### 3.5. Relative Abundance Changes of Lc and Gs in the Crucian Carp Gastrointestine

Gastrointestinal content samples, collected at five time points during the three periods (from day −7 to day 21), were analyzed using a 16S RNA gene sequencing technique, and the results are shown in Figure 4. Lc was detected at very low abundance in the gastrointestine before the administration of the mixed diet (Day–7). It is not surprising that Lc became the major taxon in terms of abundance (36.75% ± 3.59%) after the administration of the mixed diet (day 0), whereas 7 days after the cessation of the mixed diet, the relative abundance of Lc decreased to 0.11% ± 0.03%. Fourteen days later, the relative abundance of Lc decreased to a very low level again, even lower than that of the control group (Figure 4 and Figure 5).

The relative abundance of Gs had the same trend as that of Lc (see Figure 4 and Figure 5). At day 0, the relative abundance of Gs was 36.12% ± 5.31%, which was similar to that of Lc (Figure 5), but the number of viable Gs was eight times that of Lc (Figure 6). At day 7, although the relative abundance of Lc was 0.11% ± 0.03%, which was higher than other time points (except day 0), there was no viable Lc in the GI tract. We speculate that inactive Lc have reentered the GI tract because of the first incomplete replacement of the rearing water, and the same issue might also exist with the Gs. Viable Gs was detectable up to day 7, which is consistent with the results in Experiment 1. Regarding the control group, the relative Lc and Gs abundance remained at a very low level during the whole experiment, and no viable Lc and Gs were detected.

## 4. Discussion

Here, an improved, highly sensitive selective culture method was used to monitor Lc in the GI tract of crucian carp whereby interference from nontarget bacteria was eliminated. Meanwhile, a transit marker was used to assess Lc colonization. In addition, a high-throughput sequencing technique was used to further understand changes in the relative abundance of Lc and Gs.

### 4.1. Elimination Interference Is Essential for Colonization

Compared with terrestrial animals and humans, the intestinal microbiota of fish is more easily affected by feed and rearing water [3,28], Moreover, it is inevitable that there will be *Lactobacillus* and *Bacillus* in fish diet. *Lactobacillus* and *Carnobacterium* could be detected in the gut of control groups in a probiotic feeding trial [13]. Merrifield et al. [29,30] also reported that *Enterococcus* and *Bacillus* could be detected in the gut of rainbow trout that were fed a diet without probiotic supplementation, and they considered that these bacteria may be indigenous species. In the five commercial feeds, we detected different species of LAB and *Bacillus* at different concentrations. One of the feeds contained *Pediococcus* at 1.4 × 10^4^ cfu/g, and another feed contained *Bacillus* at over 10^6^ cfu/g (Table A2). Therefore, we proposed that sterile aquafeed should be used in GI microbe-related experiments. We therefore selected ^60^Co irradiation, which is a good sterilization method recommended for its wide use in SPF animal feed [20].

In the experiment, the target bacteria were more likely to reenter the gut via residual diet or feces. Merrifield et al. [29] found that 7.4 × 10^3^ cfu/mL of *Bacillus* and 4.3 × 10^3^ cfu/mL of *Enterococcus* were detected in the rearing water after feeding the diet supplemented with these bacteria, despite 15% water renewal per 72 h. Therefore, the authors suggested enhancing the water renewal rate to reduce background interference [29,30].

In rearing water with a pH of 8.0–9.0, the concentration of the Lc decreased dramatically from 1.0 × 10^6^ cfu/mL at the beginning to <1 cfu/mL 7 days later (unpublished data). Considering their short life in water, 100% water renewal with an interval of 5 days is enough to control the amount of these Lc in the water. However, if a testing strain can endure the water environment (such as in the case of a Gs spore) or even proliferate, the persistence time would be overestimated, and the reintroduction of the testing strain would be obvious. Thus, a better method for controlling the testing strain in water is needed.

### 4.2. The Improved, Highly Sensitive Selective Culture Combined with a Transit Marker Is a Suitable Method for the Study of Colonization in Fish

Various methods have been developed to evaluate bacterial colonization in complex gut microbiota. Although tagging probiotic strains with fluorescence markers is an alternative, frequent plasmid loss during gut transition, low detection sensitivity and safety concerns hinder its further application. Species-specific PCR has also been developed to directly detect organisms in the extracted genome of fecal or GI tract samples. However, it cannot eliminate the baseline values of indigenous bacteria of the same species in their environments or diets [31]. At present, strain-specific PCR is used to detect and quantify strains; however, these strain-specific DNA fragments are based on a limited number of strains, making the strain-specificity robust only within a narrow confidence interval. These methods focus on humans and mice and are not suitable for colonization studies of aquatic animals such as fish. Although a selective medium method with colony identification is considered arduous and time-consuming, it is still a classic method in microbiology studies [32]. In particular, the method can tell whether the bacteria are alive or dead, whereas molecular methods cannot.

The MRS agar with a pH of 5.4–5.5 had high specificity and sensitivity for detecting acid-resistant bacterial species in the GI tract, such as the Lc strains used in our study. The weight of GI tract samples usually does not exceed 1 g after an appropriate starvation period when the bodyweight of the fish is less than 30 g. Then, a 10% homogenate of less than 10 mL can be entirely spread on agar on fewer than 50 plates at 200 μL/plate. The detection limit using this approach is 1 cfu/gastrointestine. Other culture-dependent methods have poor accuracy and a detection limit usually higher than 10 cfu/g [13,15], whereas our improved selective culture method is very suitable for fish colonization experiments.

Colonization was defined by Conway and Cohen as the indefinite persistence of a particular bacterial population without the reintroduction of that bacterium [18]. If a microbe can exit the GI tract in the extreme long term (such as its whole life) or extreme short term (such as a couple of days), then the conclusion of colonization is not easy to make. However, if a microbe merely exits the GI tract for “a period of time”, how should we define the length of that time? Marteau and Vesa [21] indicated that using a transit marker is necessary when studying the colonization of potential probiotics, and the colonizer should persist for a longer period than the marker. A Gs spore is a good transit marker [21,25,26] for the following reasons: Firstly, its growing temperature ranges from 40 to 70 °C [33], so it usually cannot germinate, grow or reproduce in rearing water and fish gut. Secondly, the spores cannot be easily destroyed in the GI tract and feed preparation process. Thirdly, the spores can easily be counted based on high-temperature selective culture where other gastrointestinal bacteria usually cannot grow. Our study showed that the detection limit of Gs can reach 1 cfu/gastrointestine.

### 4.3. Monitored Relative Abundance Changes by High-Throughput Sequencing

With the second-generation sequencing technique for gut microbiome community analysis, we can identify bacterial components at the genus level. Some researchers employed 16S rRNA amplicon sequencing to study colonization [34,35]. Howitt compared traditional microbiological cultures and 16S polymerase chain reaction analyses for the identification of preoperative airway colonization in patients undergoing lung resection. The results showed that 16S PCR analyses identify colonizing bacteria in a similar proportion of preoperative BAL samples as traditional cultures [36]. An approach based on Illumina HiSeq 16S rRNA amplicon was used by Xia et al. [11], with results showing that *Lactococcus lactis* JCM5805 was below the detection level after the cessation of probiotics for 5 days, and they inferred that this strain could not colonize the gut; rather, the evaluation of colonization based on the 16S rRNA amplicon technology that they used is limited, for two reasons. First, the detection level of the method on a fish’s gastrointestinal sample is unknown. Metagenomics is only able to distinguish bacteria with concentrations greater than 10^6^ bacteria per gram of feces [37]; thus, some low-abundance bacteria would be missed by metagenomic analysis. Second, the method is based on DNA samples and cannot determine the viability of bacteria, i.e., whether the bacteria are alive or dead, which could influence the interpretation of the results [2]. Of course, this method is feasible as an auxiliary means to understand changes in the abundance of the target bacteria.

### 4.4. Lc ATCC 393 Cannot Colonize the Gastrointestinal Tract

The persistence of probiotics in the gut is species-specific. In our previous study, even though an exogenous *Bacillus licheniformis* A1(Bli-A1) supplement was withdrawn, the concentration of Bli-A1 in the intestinal content was sustained at 3.3 × 10^2^ cfu/g for at least 42 days with continuous sterile feed supplements [38]. In this study, when the detection limit was 1 cfu/gastrointestine, the elimination speed of Lc was even faster than that of the transit marker, indicating that Lc could not colonize in the gastrointestine of crucian carp. This is consistent with our previous studies of Lc on catfish [27]. We speculate that there are three reasons that Lc could not colonize in the gastrointestine of crucian carp. First, indigenous microbiomes drive colonization resistance to probiotics and/or additional bacteria [39]. Second, Gastrointestinal contents are not conducive to Lc reproduction. Third, Lc lacks the ability to adhere to the mucosa of the GI tract of crucian carp.

However, the supplement of Lc changed the gastrointestinal microbiota structure of crucian carp (Appendix A), compared with day −7, the number of the high-abundant taxa (≥1%) increased from 9 (except other bacteria abundance) to 24 (except other bacteria abundance) on day 7, and recovered to the previous (day −7) microbiota structure until day 21.

## 5. Conclusions

The elimination speed of Lc was faster than the transit marker. Meanwhile, although Lc retained a low relative abundance from day 7 (0.11% ± 0.03%) to 21 in the crucian carp gastrointestine, they were not viable. The results indicate that the Lc ATCC 393 cannot colonize crucian carp. This study presents a method with a low detection limit for the colonization of LAB in fish and provides the idea of crucian carp to screen for beneficial probiotics.

## Figures and Tables

**Figure 1 microorganisms-09-02547-f001:**
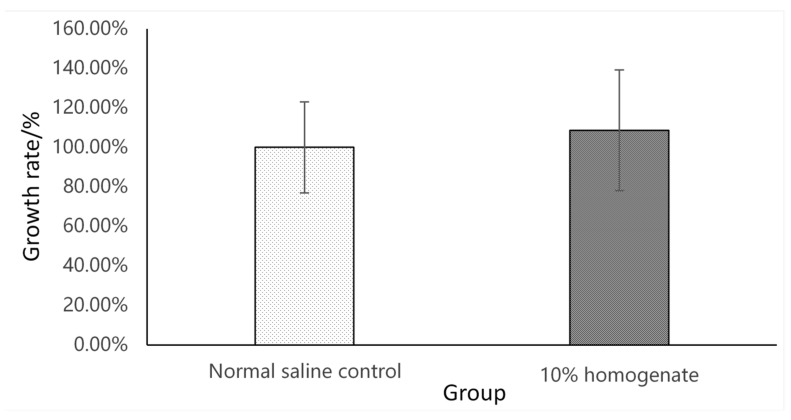
Comparison between the growth rate of *L. casei* in the normal saline control and 10% GI tract homogenate (*n* = 9) on the MRS plate.

**Figure 2 microorganisms-09-02547-f002:**
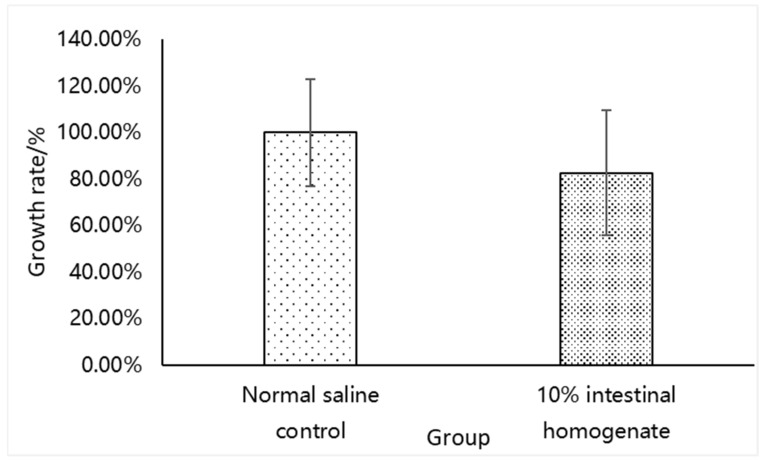
Comparison between the growth rate of Gs in the normal saline control and 10% GI tract homogenate (*n* = 9) on the NA plate.

**Figure 3 microorganisms-09-02547-f003:**
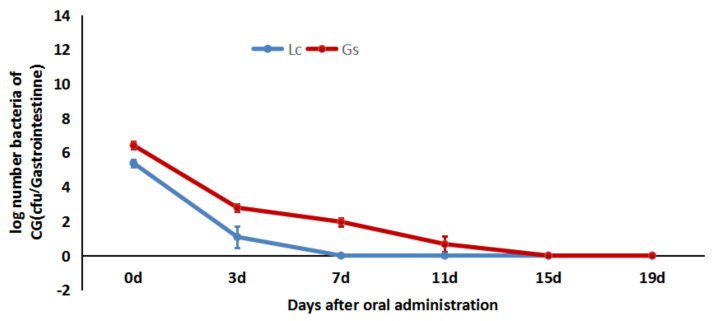
Kinetics of Lc and Gs elimination in the GI tract of crucian carp (*n* = 9).

**Figure 4 microorganisms-09-02547-f004:**
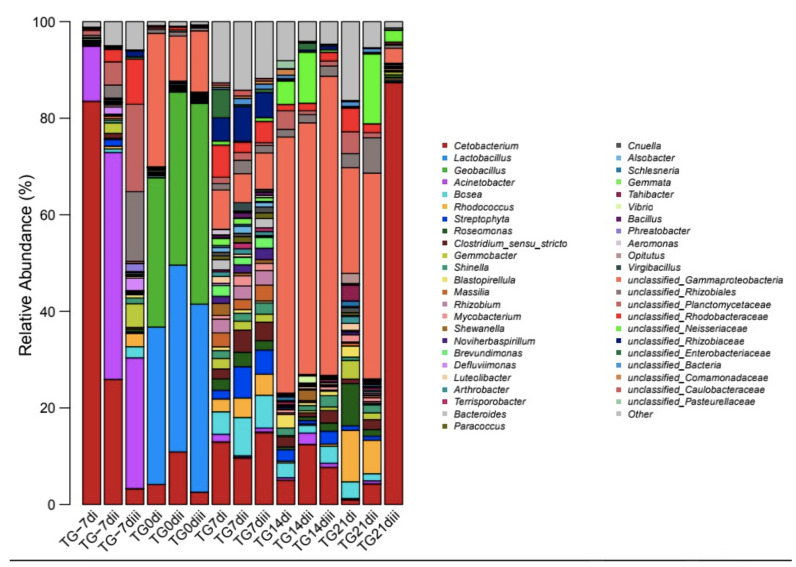
Bar plot illustrating the relative higher abundance bacterial genera for the individual fish. TG: treatment group: −7, 0, 7, 14 and 21 d represent the sample time points; i, ii, and iii represent individual triplicates within a group.

**Figure 5 microorganisms-09-02547-f005:**
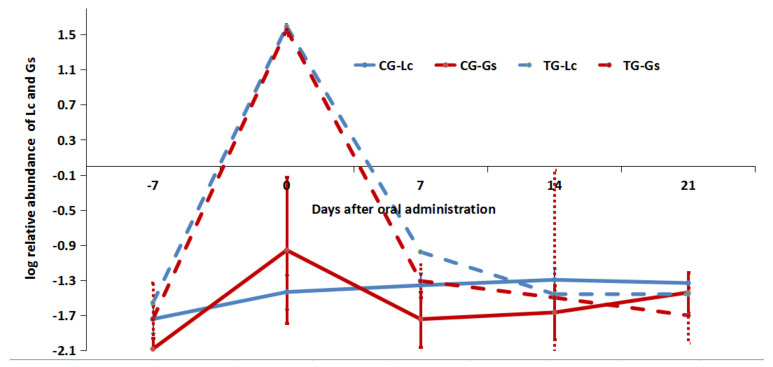
The changes in relative abundance of Lc and Gs in the CG and TG from day −7 to 21.

**Figure 6 microorganisms-09-02547-f006:**
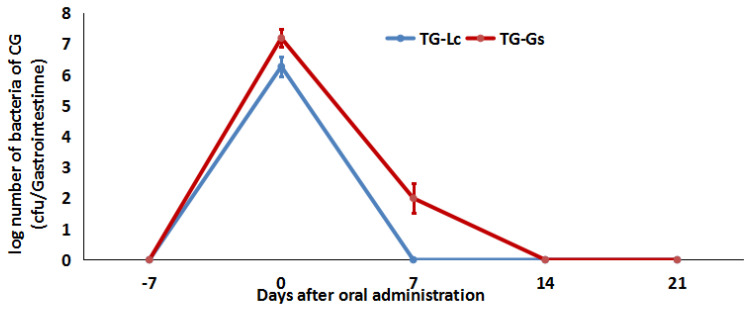
The changes in viable Lc and Gs bacteria in the TG from day −7 to 21.

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
