# Peer review of "Lacticaseibacillus casei* ATCC 393 Cannot Colonize the Gastrointestinal Tract of Crucian Carp"

_microorganisms, 2021, doi:10.3390/microorganisms9122547_

Round 1
Reviewer 1 Report
The manuscript need a great reorganization in material and methods and discussion sections.
Also, introduction must be rewritten for a deeper introduction about the state of the art and the objectives to achieve in this research.
Finally, in the conclusions section, an effort should be made to further develop the written paragraph.
After making these modifications, it will be my pleasure to enter into an in-depth discussion about the results obtained.
Author Response
Thanks for your advice. We have rewritten some parts including abstract and conclusion, and revised other places, see the revised manuscript. And our manuscript have received the specialist edit service in MDPI.
Reviewer 2 Report
The authors show that the potential probiotic bacterial species Lacticaseibacillus casei does not colonize the gut of the common carp.
The primary issues with this manuscript stem from a need for substantial language editing before publication and some clarification in the methods.
One concern with the relevance of this manuscript is that probiotics may have beneficial effects, even if they do not colonize the intestinal tract. There was no measurement or report here of the effects of Lc on these fish, which leads me to wonder, why is its colonization being tested, and what beneficial properties it may have to the carp.
Another concern is that the authors do not address why they chose Geobacillus stearothermophilus as their transit marker, rather than another bacterial species or even something non-biological. The authors also need to make more explicit how they know that Gs spores were introduced to the fish guts rather than vegetative cells.
This manuscript needs to be edited for spelling and syntax
Reviewer 3 Report
The paper entitled ‘Lacticaseibacillus casei cannot colonize the gastrointestinal 2 tract of Crucian carp’ written by Zhang et al. presented thorough investigation of bacterial community and colonization mechanism of L. casei in the tract digestive of Crucian carp. Although the article needs deep English check, it is well structured and systematic. The author analyzed colonization mechanism employing two bacteria as a marker and target, which are L. casei and G. stearothermophilus, respectively. In addition, they also reported relative abundances of bacterial community compositions in tract digestive samples over several sampling points.
Despite interesting findings, the authors still need discuss essential topic such as interaction between L. casei, G. stearothermophilus, and other bacteria in the tract digestive, whether they are independent each other, having mutualistic interaction or conversely, they compete and inhibit each other. This topic has not been deeply discussed yet. Moreover, if the L. casei cannot colonize the tract digestive, what will the implication be in term of colonization mechanism study, crucian carp aquaculture perspective and what the author could suggest to improve the colonization.
The author shall also mention animal welfare status once they did their experiment. This can be done for example in the conflict of interest or in material and methods section informing that the authors performed research according to an approved/qualified animal welfare for research method.
Overall, the manuscript is suitable for publication in mdpi journal, chapter Microorganisms after major revision. Detail comments are mentioned below.
Abstract
- Sentences in abstract section contain grammar mistake. Please revised accordingly (line 16, 22, 24-25).
- Line 26-27, high-throughput sequencing combined with SCM was used to reconfirm the elimination kinetic of Lc in GI tract. Is this a method or result?
-If there is no colonization, how can we reply on selective culture combined with transit marker method as a method for colonization study?
Material and methods
Line 81: MRS abbreviation?
Line 82-83: How much is the needed concentration?
Line 87-88: How much is the needed concentration?
Line 98-103: Why did the concentration of Lc and Gs differ and why the Lc concentration was higher than Gs?
Line 104: Typing error: raring --> rearing. Please apply to other similar words.
Line 104: Typing error: raring --> rearing. Please apply to other similar words.
Line 135-136: rephrase, the colonies were randomly chosen for 16S rRNA.
Line 143: What is the composition of normal saline solution?
Line 146 and 153, Why did you use 10% intestine homogenate?
Line 167: How and why did you make threshold of detection limit to 2 cfu/intestine?
Results
Line 189: What is the thermophiles bacteria you meant here?
Line 198-201: Pantoea et al? What is this?
Line 201-203. Why the thermophiles were present in the fish intestine?
Discussion
Line 265: L and B -> in capital.
Line 275: do you think that the fish ate their feces?
Line 286: Could you give an example or suggestion about a better method?
Line 306-310. The authors need to give reference why they used 10% of intestinal homogenate. I would argue that if the nutrient from intestinal homogenate is to little/insufficient, targeted bacteria may not grow.
Line 324-341. The authors seemed to be confused with 16S rRNA amplicon sequencing. This method is not specifically for bacterial colonization, but it aims to identify taxonomic identity of bacteria or archaea. Moreover, the cited reference (line 337 (ref.no. 36) and in line 340 (ref.no.2) did not mention the phrase as of the authors wrote.
Line 334: What is exogenous Bli-A1?
Line 350-352. What is the indigenous microbe in crucian carp? are they different from catfish? How do you support your second and third hypothesis that intestinal content is not conducive and that Lc lacks ability to adhere crucian carp's tract?
Conclusion
Line 354-358. The conclusion is unclear. Based on 16S rRNA profile this bacteria was in low abundance in almost samples. In addition, they have similar trend as of Lc. Authors may also give what is the benefit of their study for colonization bacteria topic as well as possible application of their study for aquaculture of crucian carp.

Round 2
Reviewer 1 Report
Summary
The MS titled “Lacticaseibacillus casei cannot colonize the gastrointestinal tract of crucian carp” presents a study about the capacity of L. case of colonize gastrointestinal tract of crucial carp using a microorganism as marker (Geobacillus stearothermophilus).
Authors conclude that the Lc is unable to colonize the crucial carp GI.
Reading the MS has been incommodious and I disagree with most of the statements:
- During the MS authors state “The therapeutic benefits of probiotics are that they can colonize intestinal mucosa and thereby modulate the intestinal flora(*)”, . Most experts in microbiota, and especially fish microbiota, are not entirely sure that probiotics must colonize the GI to exert their effect.
Gastro intestinal colonization is not a benefit per se, it is believed that is one of the mechanisms by pathogens colonization is inhibited There are a few very superficial studies where this fact has been described indicating that is rarer than we think. Actually, some authors would rather use the term “temporal colonization” than “colonization” when discussing about probiotic persistence and presence within the intestinal mucosa [1].
(*) the term "intestinal flora" now a days is not used, instead is recommended to use "intestinal microbiota".
- It is very important to know the microorganisms’ strains, because not all species can be used as fish probiotics. So, when in lines 51-54 bacterial speciess are mentioned also their strains must be included.
- In lines 78 – 80 is stated that “once the probiotic supplementation has ceased, the proportion of the target strain may remain at very low level, requiring a detection method with higher sensitivity”. This method already exists yet, an authors used it in their study, metabarcoding. So, this is not a novelty.
- The authors used the Lacticaseibacillus casei ATCC 393 strain for the probiotic colonization study. In the title and the conclusions section they state that this species (Lactocaseibacillus casei) cannot colonize the GI of crucian carp. This is a very risky sentence to affirm, since it is known that the probiotic effect and therefore the colonization, is a dependent characteristic of the strain. Maybe in the MS authors must write the strain used.
Furthermore, no study has previously been conducted on the resistance of this strain to gastrointestinal conditions. Perhaps the low number of Lc. casei that is found in the intestine is due to the fact that it does not resist the pH of the stomach, nor the digestive enzymes, nor the bile salts, for example.
[1] X. Li, E. Ringø, S. H. Hoseinifar, H. L. Lauzon, H. Birkbeck, and D. Yang, The adherence and colonization of microorganisms in fish gastrointestinal tract, vol. 11, no. 3. Wiley-Blackwell, 2019, pp. 603–618. doi: 10.1111/raq.12248.
Herein are other changes that I recommend to make:
Materials and methods
- In all section there is a lack of the brands of the reactants and culture mediums used.
- In lines 96 – 100 authors explain how Geobacillus stearothermophilus is grown before to be added to aquafeed. But in line 120 it is said that “a transit marker (a GS spore) is used”. Authors must clarify this point.
- Also, what “normal saline” means. Please said what salt is used and solution pH.
- In line 102, it is said the “aquafeeds were sterilized by Co60 radiation”, but results showed in table A1 contradict it. It should be more appropriated say that the bacterial load is reduced at to a percentage. So initial microbial load of the aquafeeds must be analysed. Also it must be respecified what commercial aquafeed is used.
- In line 110, it is said that diet 2 was used in the experiment. Why this diet was chosen must be explained.
- Again, it is very difficult to understand the experiments sections. Maybe a figure can help to follow the text, and clarify all the procedure. For example, what experiment 1 and 2 are; but if I have understanded well this section, I think is better to say Analysis methodology 1 and 2.
- Colonies identification by 16S rRNA gene sequencing must be explained. From DNA extraction to database used for DNA alienation and strain identification.
- I think that the way that cfu/ml are counted is not correct. The procedure broadly accepted for aerobic and facultative aerobic microorganism is the serial dilution method. This method also let to make a statistical study, and state that the detection limit of a method is 1 or 2 cfu/ml is not right, this limit must be accompanied with a standard deviation.
- In lines 157-158 it is said the entire GI tract was removed from the esophagus to anus. If GI homogenate is obtained from this section, not represent GI conditions. I think that to make the homogenate only the intestine section is used. In that case this fact must be explained.
- The title of 2.6 section must reflect the method used. For example: “16S rRNA metabarcoding”.
- Also, is section 2.6.1. primers used must be indicated.
Results
- In my opinion, Figures 1 and 2 can be omitted if their data is written in the text.
- And how the growth rate is calculated must be explained in methods section.
- In Figure 4 caption it is written that there are illustrated “various bacterial genera”. The term “various” it is not appropriate in this context. It should be clarified whether are represented the most abundant or important genera, for example.
Discussion
- In lines 290 and 349 “aquaculture water” must be changed by “rearing water” which is more appropriated.
- In lines 299-300, authors write that they “selected 60Co (*) irradiation because is a good sterilization method recommended and it is widely uses in SPF feed of experimental animals. The study of the reference which the authors mention (19), it is used a dose of 12 kGy which sterilize the animal’s feed. Why this dose is not used in this experiment? In fact, there is not said what dose is used with the aquafeeds in this study.
(*) Write correctly this chemical isotope.
- Finally, and as I said before, I do not agree with the phrase "Lc cannot colonize the GI". And also, speculations as to why this species cannot colonize the gastrointestinal tract are just a few of the many that exist or are thought to exist. First, perhaps it should be said that researchers have speculated about why a microorganism cannot colonize the gastrointestinal tract of a fish. And second, more thorough bibliographic research must be carried out on the reasons why this occurs, because there are more than the three exposed.

Reviewer 2 Report
The authors have adequately addressed my concerns for this manuscript
Author Response
thanks.
Reviewer 3 Report
The authors revised their manuscript according to reviewer's comment. In addition, they also answered questions and supported their argument with cited references. The manuscript is considered for a publication in the MDPI journal, chapter Microorganisms.
Author Response
thanks.